

# Simulating the Influence of Primary Biological Aerosol Particles on Clouds by Heterogeneous Ice Nucleation

Matthias Hummel[1,*], Corinna Hoose[1], Bernhard Pummer[2], Caroline Schaupp[1], Janine Fröhlich-Nowoisky[2], and Ottmar Möhler[1]

[1]Institute of Meteorology and Climate Research, Karlsruhe Institute of Technology, Karlsruhe, Germany
[2]Department of Multiphase Chemistry, Max Planck Institute for Chemistry, Mainz, Germany
[*]now at: Department of Geosciences, University of Oslo, Norway

*Correspondence to:* Matthias Hummel (hummel@geo.uio.no)

**Abstract.** Primary ice formation, which is an important process for mixed-phase clouds with impact on their lifetime, radiative balance and hence the climate, strongly depends on the availability of ice nucleating particles (INPs). Supercooled droplets within these clouds remain liquid until an INP immersed in or colliding with to the droplet gets reaches its activation temperature. Only a few

aerosol particles are acting as INPs and the freezing efficiency varies among them. Thus, the fraction of supercooled water in the cloud depends on the specific properties and concentrations of the INPs. Primary biological aerosol particles (PBAPs) have been identified as very efficient INPs at high subzero temperatures, but their very low atmospheric concentrations make it difficult to quantify their impact on clouds.

Here we use the regional atmospheric model COSMO-ART to simulate the heterogeneous ice nucleation by PBAPs during a 1-week case study on a domain covering Europe. We focus on three highly ice nucleation active PBAP species, *Pseudomonas syringae* bacteria cells and spores from the fungi *Cladosporium* sp. and *Mortierella alpina*. PBAP emissions are parameterized in order to represent the entirety of bacteria and fungal spores in the atmosphere. Thus, only parts of the

simulated PBAP are assumed to act as INP. The ice nucleation parameterizations are specific for the three selected species and are based on a deterministic approach. The PBAP concentrations simulated in this study are within the range of previously reported results from other modelling studies and atmospheric measurements. Two regimes of PBAP INP concentrations are identified: a temperature-limited and a PBAP-limited regime, which occur at temperatures above and below

a maximal concentration at around $-10°C$, respectively. In an ensemble of control and disturbed simulations, the change in the average ice crystal concentration by biological INPs is not statistically



significant, suggesting that PBAP have no significant influence on the average state of the cloud ice phase. However, if the cloud top temperature is below $-15°C$, PBAP can influence the cloud ice phase and produce ice crystals in the absence of other INPs. Nevertheless, the number of produced ice crystals is very low and it has no influence on the modelled number of cloud droplets and hence the cloud structure.

## 1 Introduction

The initial formation of ice crystals in mixed phase clouds is catalyzed by ice nucleating particles (INP) at certain temperatures and relative humidities, depending on the properties of the aerosol particle acting as INP (Hoose and Möhler, 2012). It is the current consensus that only a small proportion of all atmospheric aerosol particles can act as INPs. Among these are certain types of mineral dust, metallic and combustion particles (Murray et al., 2012). Some primary biological aerosol particles (PBAP) are found to be ice nucleation active in the immersion freezing mode, i.e. initiating freezing from the inside of a supercooled liquid cloud droplet. When comparing the freezing onset temperatures, some ice nucleation active bacteria and fungi are among the most ice nucleation active particles present in the Earth's atmosphere (Després et al., 2012; Hoose and Möhler, 2012; Murray et al., 2012; Morris et al., 2013).

PBAP are an ubiquitous group of the Earth's atmospheric aerosols and are defined as solid and insoluble particles of biological origin (Elbert et al., 2007; Després et al., 2012; Fröhlich-Nowoisky et al., 2016). PBAP include microorganisms (e.g. bacteria) as well as reproductive units from the biosphere (e.g. fungal spores, plant pollen) (Després et al., 2012). Single bacteria cells and agglomerates have the size of typically $0.25-3\,\mu m$ and $3-8\,\mu m$ in diameter, respectively, whereas actively wet discharged ascospores[1] are usually $2-20\,\mu m$ and actively wet discharged basidiospores[1] are usually $1-10\,\mu m$ in aerodynamic diameter (Gregory, 1961; Elbert et al., 2007).

Ice nucleation active PBAP potentially influence cloud microphysical properties by increasing ice formation at high subzero temperatures (e.g., DeMott and Prenni, 2010), and subsequently also impact cloud dynamics. It has been hypothesized that a more effective glaciation of cloud droplets decreases the cloud lifetime and the cloud albedo (termed glaciation indirect effect when referring to anthropogenic INP (Lohmann and Hoose, 2009; Boucher et al., 2013). The presence of specific INP in clouds can be observed indirectly by analyzing cloud and precipitation particle residuals (Cziczo et al., 2013). Biological particles have been observed ubiquitously in precipitation and snowpack at ground level (e.g. Christner et al., 2008; Stopelli et al., 2015), inside clouds by airborne measurements (e.g. Pratt et al., 2009; DeLeon-Rodriguez et al., 2013), and in cloud water collected at mountain top stations (Joly et al., 2014). Furthermore, they have been shown to significantly con-

---

[1]Spores of fungi from the phyla *Ascomycota* or *Basidiomycota*, respectively.


tribute to INP concentrations measured in cloud-free air at ground level (Huffman et al., 2013; Prenni
et al., 2013; Creamean et al., 2013).

Previous modelling studies suggest that primary biological INP (PB-INP) have a limited influ-
ence on clouds and precipitation on the global average, because the INP number concentrations of
mineral dust and soot are found to be several orders of magnitude higher than the INP number con-

centration of PBAP (Hoose et al., 2010b; Sesartic et al., 2012, 2013; Spracklen and Heald, 2013).
However, these studies neglect constraints on the ice nucleation activity of non-biological particles
at high subzero temperatures ($T > -15°C$), where DeMott and Prenni (2010) suggest that PB-INP
are most important for ice nucleation in mixed-phase clouds. In this range, laboratory measurements
of the ice nucleation activity of mineral dust and soot are scarce, since nucleation rates are too low

at high temperatures to be measured in the laboratory under representative conditions. However,
the parameterizations used in previous model studies are extrapolated from measurements at lower
temperatures. In contrast, most laboratory measurements of bacteria INP are reported in this temper-
ature range. Different parameterization approaches for these have been suggested and strongly im-
pact model results (Sahyoun et al., 2016). Furthermore, in parameterizations based on field studies

(e.g., Phillips et al., 2008), the attribution of the INP activity to different aerosol types is ambigu-
ous. Secondary ice formation e.g. from splinter production during ice-droplet collisions (Hallett and
Mossop, 1974) has been suggested to be active at $T > -8°C$ and could enhance the effect of biolog-
ical ice nucleation. The model representation of these processes is uncertain. In regimes colder than
$-15°C$, mineral dust and other non-biological INPs are also active and the relative abundance of

PBAP compared to inorganic particles is probably too small for PB-INP to contribute significantly
to ice formation. With the coarse horizontal and vertical resolution employed in global models, this
narrow temperature window, in which PB-INP are thought to be most effective, is poorly resolved.
Also, local concentration maxima of biological particles close to source areas can not be captured
by global models.

In this study, we present mesoscale model simulations of PBAP and their interactions with clouds
with a grid spacing of $14\ km$. For this, a limited-area atmospheric model is extended by surface
emissions, atmospheric dispersion, and heterogeneous ice nucleation for two different PBAP (bacte-
ria and fungal spores). Recent laboratory measurements of the ice nucleating efficiency of different
PBAP are used to derive parameterizations for immersion freezing. The number concentrations of

PB-INP during a 1-week episode in July 2010 in Europe are analysed and compared to background
dust INP concentrations. Furthermore, the impact of PB-INP on clouds is quantified in ensemble
simulations with and without PBAP acting as INP.



## 2 Methodology

### 2.1 Emission and Dispersal of Biological Particles

The COSMO-ART (Consortium for Small-Scale Modelling - Aerosols and Reactive Trace Gases) regional atmospheric model system is based on the forecast model of the German weather service, combined with an two-way interaction module for simulating the spatial and temporal distribution of reactive gaseous and particulate components (Vogel et al., 2009). In this study, the treatment of two types of PBAP has been included in the model. Bacteria and fungal spores are chosen because

of their significant contributions to the PBAP mass and number (Després et al., 2012) and because highly ice nucleation active species have been identified within these groups (e.g., Maki et al. 1974 for *Pseudomonas syringae*; Fröhlich-Nowoisky et al. 2015 for *Mortierella alpina*). Pollen grains can also be simulated with COSMO-ART (Vogel et al., 2008; Zink et al., 2013). However, as their concentrations are significantly lower than bacteria and fungal spore concentrations, they are not

considered any further in this work.

The physical properties of the simulated PBAP are listed in Table 1. Bacteria and fungal spores are assumed to be monodisperse and spherical, except for the calculation of the sedimentation velocity, which assumes prolate spheroids (Hummel et al., 2015).

**Table 1.** Physical characteristics of the different PBAP types used for the model simulation of this study. $a$ and $b$ are the semi-axes of the prolate spheroid. [1] Hinds 1999, Lamanna et al., 1973; [2] Schaupp, 2013; [3] Hummel et al. (2015); [4] Haga et al. (2013)

| | $d_P (\mu m)$ | a $(\mu m)$ | b $(\mu m)$ | $\rho_P (kg/m^3)$ |
|---|---|---|---|---|
| bacteria (*Pseudomonas syringae*) | $0.6^{[2]}$ | 0.55 | 0.65 | $1100^{[1]}$ |
| fungal spores (*Cladosporium* sp., *Mortierella alpina*) | $3^{[3]}$ | 1 | 5 | $1000^{[4]}$ |

PBAP are treated as inert tracers, neglecting any interactions with other aerosols or gases (coagu-

lation or condensation), as well as fragmentation and bursting processes. The temporal development of the PBAP number concentration is calculated by the following prognostic equation:

$$\rho \frac{d\Psi}{dt} = -\nabla \bullet \boldsymbol{F_T} - \frac{\partial}{\partial z}F_S - \lambda \, \Psi - \frac{1}{N} \, \frac{\partial}{\partial z}F_E \qquad (1)$$

with the number mixing ratio of PBAP $\Psi = \frac{N_f}{N}$. Here, $N_f$ is the number concentration of PBAP, $N$ is the total number of particles and air molecules per volume of air, $\rho$ is the air density, $\boldsymbol{F_T}$ is

the turbulent flux, $F_S$ is the sedimentation flux, $\lambda$ is the washout coefficient, and $F_E$ is the vertical emission flux (Vogel et al., 2008; Helbig et al., 2004). $F_E$ is calculated separately for each type of PBAP by means of an emission parameterization: $F_{E,B}$ for bacteria and $F_{E,FSP}$ for fungal spores.

$F_{E,B}$ is the total emission flux of bacteria to the atmosphere. It consists of the sum of constant fluxes specified for particular ecosystems (Burrows et al., 2009). Each ecosystem specific flux has

been derived by adapting simulated bacteria concentrations to the observed near-surface bacteria





number concentrations (Burrows et al., 2009). Therefore, the total bacteria emission flux is given by the sum of all individual emission fluxes multiplied by the tile fractions of the respective ecosystems (Hoose et al., 2010a). Descriptions of the ecosystem coverage used for the used model simulation are given in section 2.4.

The fungal spore emission is described by a flux $F_{E,S}$ depending on selected meteorological and ground parameters as derived from a previous model study with COSMO-ART (Hummel et al., 2015). In Hummel et al. (2015), the simulated fungal spore concentration has been adapted to local near-surface concentrations of measured fluorescent biological aerosol particles (FBAP) in Central and Northern Europe. It is assumed that fungal spore concentrations may be best approximated

by FBAP concentrations as the dominant FBAP size mode frequently coincides with typical fungal spore sizes (Pöschl et al., 2010; Huffman et al., 2012; Hummel et al., 2015). The FBAP emission flux has been fitted to measured FBAP concentrations with the assumption that the biological particles are evenly distributed throughout the boundary layer. As a result, the emission flux in $m^{-2}s^{-1}$ is given by the following function:

$$F_{E,S} = b_1(T - 275.82K) + b_2\, q_v\, LAI \qquad (2)$$

where $T$ is the surface temperature in $K$, $q_v$ the specific humidity in $kg kg^{-1}$, $LAI$ the leaf area index in $m^2 m^{-2}$, and $b_1 = 20.426$ and $b_2 = 3.93 \times 10^4$ are fit parameters (Hummel et al., 2015).

### 2.2 Parameterization of Heterogeneous Ice Nucleation

Heterogeneous freezing of droplets containing an immersed PBAP is assumed to occur on specific
ice nucleation active surface sites (INAS) at a characteristc temperature (Pruppacher and Klett, 1997; Connolly et al., 2009). For this approach, the surface density of ice nucleation active sites is labeled $n_S(T)$ and describes the number of surface sites that are ice nucleation active between $0°C$ and $T$ (Connolly et al., 2009). For monodisperse particles, the ratio $f_{IN}$ of PB-INP to total PBAP is then a function of $n_S(T)$ and the individual particle surface area $A_P$ calculated by:

$$f_{IN}(T) = 1 - \exp(-A_P\, n_S(T)) \qquad (3)$$

    $f_{IN}(T)$ describes a functional form with a slope depending on $n_S(T)$ towards higher temperatures and which stays constant towards lower temperatures if $f_{IN} = 1$. The latter implies that every particle gets activated as INP, which is not the case for some PB-INP (Figure 1b). Hence, $f_{IN}$ is modified by $\widetilde{f}_{IN}(T) = \gamma f_{IN}(T)$, where $\gamma \leqslant 1$ describes the constant value when every potentially
ice nucleation active PBAP is activated as INP. Some PBAP of the same species can remain non-activated, and $\gamma$ can be estimated from the low-temperature tail of the laboratory results described here, where $\widetilde{f}_{IN}(T)$ is approximately constant with $T$.

    Taking this modification into account, a temperature dependent INAS density is derived from different laboratory experiments. *Pseudomonas syringae* bacteria have been investigated at the AIDA



cloud simulation chamber (Schaupp, 2013; Amato et al., 2015), data for *Cladosporium* sp. spores are derived with a flow cell with defined relative humidity and temperature (Iannone et al., 2011), and results for *Mortierella alpina* have been obtained by analyzing the washing water from the mycelium, which can contain spores (Fröhlich-Nowoisky et al., 2015). The measurements are also shown in Fig. 1. By least-square fitting, the parameters for the $n_S$ temperature spectrum can be

derived:

$$n_S(T) = \exp(\alpha(T - 273.15) + \beta) \quad \text{for} \quad T < T_{max} \tag{4}$$

including temperature $T$ in $K$ and parameters $\alpha$ and $\beta$ and listed in Table 2. The parameterizations of $n_S(T)$ for different PBAP and the consistent laboratory data from which they are derived are shown in Figure 1a.

For INP descending from the species *Mortierella alpina*, we directly fit $f_{IN}$ and no $n_S$ could be calculated, because no size information was given with the measurements (Fröhlich-Nowoisky et al., 2015).

$$f_{IN}(T) = 1 - \exp(\alpha(T - 273.15 + \beta) \tag{5}$$

The upper temperature limit of the parameterization ($T_{max}$, Table 2) is defined as the highest temper-
ature at which ice nucleation with the same PBAP species as used for this study has been observed in any laboratory experiments, including other with a lower detection limit. $T_{max}$ may therefore be outside the temperature range covered by the laboratory data used for fitting the parameterization. An upper temperature limit is also given for *Mortierella alpina*, but not used in the simulation as the slope of the parameterization is very steep close to $T_{max}$ and does not require a threshold.

**Table 2.** Parameters for parameterizing the heterogeneous ice nucleating ability of different PBAP used in the model simulation of this study (Eq. 3, 4, 5, 6). [1] Amato et al. (2015); [2] Després et al. (2012); [3] Iannone et al. (2011); [4] Lang-Yona et al. (2012); [5] Fröhlich-Nowoisky et al. (2015).

| | $\alpha$ | $\beta$ | $\gamma$ | $T_{max}$ | $\varepsilon$ |
|---|---|---|---|---|---|
| Bacteria (*Pseudomonas syringae*) | -0.894 | 15.501 | 0.028 | -3°C[1] | 4%[2] |
| Fungal spores (*Cladosporium* sp.) | -0.339 | 11.567 | 1 | -28.5°C[3] | 29%[4] |
| Fungal spores (*Mortierella alpina*) | 0.37573 | 4.23229 | 0.00851 | (-5°C[5]) | 8%[5] |

The number concentration $N_{bioIN}$ of PB-INP is calculated diagnostically by applying the parameterization as a function of $T$ and PBAP number concentration $N_f$:

$$N_{bioIN} = N_f \varepsilon \widetilde{f}_{IN}(T) = N_f \varepsilon \gamma f_{IN}(T) N_{aer} \tag{6}$$



Here, $\varepsilon$ is the ratio of potentially ice nucleation active to all the PBAP species of the same type (here: bacteria or fungal spores). For example, potentially ice nucleation active bacteria are represented

in this study by the species *Pseudomonas syringae*, which can have an abundance of up to $4\%$ of all bacteria in ambient air (Després et al., 2012). Therefore, as an upper estimate, $\varepsilon$ is set to 0.04. Both types of ice-nucleation fungal spores used here also represent only a subset of the total number of fungal spores in the atmosphere. Spores of the species *Cladosporium* are among the most abundant type types of spores found in sampled air close to the ground (Lang-Yona et al.,

2012; Iannone et al., 2011). A more recent study discovered *Mortierella alpina* to be especially ice-nucleation active at high temperatures (Fröhlich-Nowoisky et al., 2015). Here, their abundance in air was estimated from the frequency of CFU (colonial forming units) found in the soil, which we assume to be fungal spores. Due to the different characteristics, both fungal spore types are included in the simulations. Contrary, most ice nucleation active bacteria show a 'Pseudomonas-

like' freezing behavior and are therefore included in the $\varepsilon$-value selected here. It should be mentioned that estimations for $\varepsilon$ have to be taken with extreme caution, since (i) the microbial diversity is huge, with many species remaining undetectable, (ii) the activity of a species depends on strain, available nutrients and growth conditions, which might be different for laboratory cultures, (iii) only few studies have been reproduced by more than one groups with different methods.

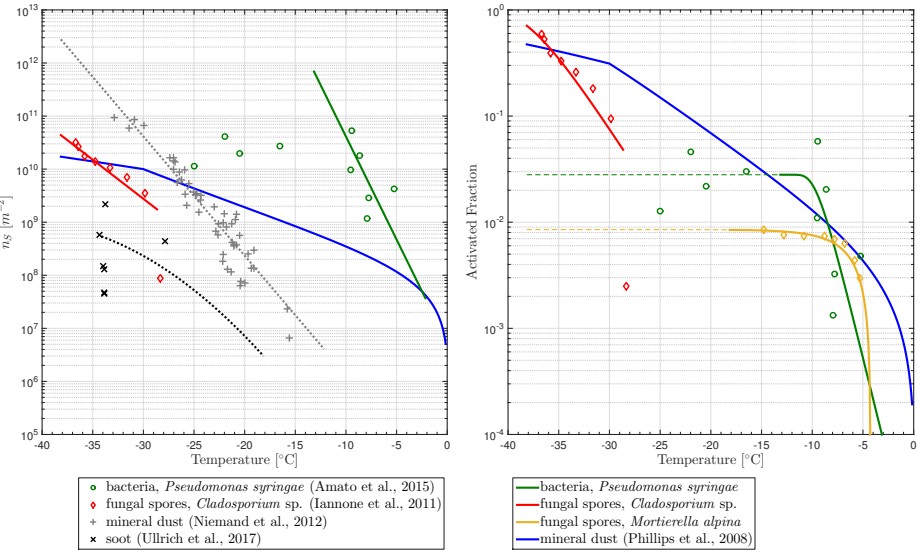

**Figure 1.** (a) INAS density $n_S$ and (b) ice active fraction $f_{IN}$ versus Temperature $T$ for PB-INP (*Pseudomonas syringae* bacteria, *Cladosporium* sp. and *Mortierella alpina* fungal spores; $r_N$ of Table 1 used here) as well as mineral dust (Phillips et al., 2008, assuming a particle size of $r_N = 3.45 \times 10^{-6}$) and soot (Ullrich et al., 2016) used in this study.





### 2.3 Cloud Microphysics

A two-moment cloud microphysics scheme is used for calculating mass and number densities of cloud and precipitation particles separately (Seifert and Beheng, 2006). The size distribution of liquid drops is split into cloud droplets and raindrops at a defined mass threshold. Warm phase processes refer to the formation and growth of cloud droplets and raindrops by nucleation, condensation, and collection of smaller droplets as well as to disruption by collisional breakup into fragments and evaporation at subsaturated regions (Seifert and Beheng, 2006). Within this chain of processes, nucleation of cloud droplets is represented by an aerosol activation parameterization depending on size distribution and chemical mixing state of aerosol particles which can potentially act as cloud condensation nuclei (CCN) (Nenes and Seinfeld, 2003). PBAP are not included in the CCN scheme because of their low number concentrations.

Within the processes including the ice phase, the scheme for nucleation of cloud ice has been modified in order to take the contribution of PB-INP into account (section 2.3). Hence, the number concentration of PB-INP is added to the total INP number concentration before calculating the ice crystal concentration.

In the original scheme, the number concentration of cloud ice crystals with a prescribed minimum mass is given by a maximum supersaturation ratio that is calculated per grid box. Heterogeneous freezing considers the size distribution and the composition of a polydisperse aerosol population (Barahona and Nenes, 2009) and uses a parameterization by Phillips et al. (2008) for mineral dust and soot INP. Insoluble organics as described by Phillips et al. (2009) potentially represent at least partly ice nucleating active bacteria, but are not included in the present model setup. Towards warmer temperatures, no upper threshold for heterogeneous freezing of mineral dust is defined in Phillips et al. (2008). In order to account for potential ice-active site contributions from embedded organic compounds in the mineral dust parameterization, an upper temperature threshold of $-15°C$ is used for non-PBAP ice nucleation (DeMott and Prenni, 2010).

### 2.4 Model Setup

The COSMO-ART mesoscale model system is driven by initial and boundary data for meteorological conditions. They are updated every six hours and result from interpolation of the coarse grid operational atmospheric model analysis of the ECMWF (European Centre for Medium-Range Weather Forecasts). No initial and boundary concentrations are predefined for aerosols or gases. Therefore, all gaseous species are set to climatological, homogeneously distributed initial concentrations. Emission rates for chemical compounds included in the ART module are updated hourly. They are provided by EMPA (Swiss Federal Laboratories for Materials Science and Technology) based on the TNO/MACC (Monitoring Atmospheric Composition and Climate) inventory (Kuenen et al., 2011). The treatment of emissions for COSMO-ART is described in Knote et al. (2011). Ho-



mogeneously distributed mass densities for each aerosol are used as initial conditions, together with initial size distributions. Primary particle emissions are parameterized based on meteorological and surface conditions. Land use data and constant surface properties are derived from the GLC2000 database (Bartholomé and Belward, 2005). All parameters are post-processed to the rotated spherical coordinate system of COSMO-ART (Doms and Schättler, 2002). For the purpose of this paper,

the model domain covers most parts of Western Europe from mainland Portugal to northern Finland, the longitudinal extension being $2849\,km$ the latitudinal extension being $3803\,km$ with a horizontal spacing of $0.125°(14\,km)$ on a rotated grid. A one-week case study in July 2010 (22 to 29 July 2010) is simulated by the model. The time of year is chosen to cover the maximal concentrations for fungal spores and bacteria within their annual cycle, as they are expected to have the largest influence on

cloud properties. In vertical direction the model reaches up to an altitude of about $24\,km$ distributed over 40 terrain-following levels. The timestepping of the Runge-Kutta dynamical core is set to $30\,s$. Model results are written to output files with a time resolution of one hour.

     Small changes in the present meteorological base state may influence the effects of PB-INP on cloud properties. Therefore, an ensemble of five independent model runs has been created by shifting

large-scale atmospheric fields in each horizontal direction (Schlüter and Schädler, 2010; Sasse and Schädler, 2014). In the present model setup, the domain is shifted by two grid boxes ($28\,km$) in each direction. Two different cases are considered for all five ensemble members, a control run without PB-INP ("DST") and a modified case including all types of PB-INP as described in section 2.3 ("BIO"). Influences of PB-INP on cloud properties are given by differences of ensemble means

between both cases (BIO - DST). Before calculating the ensemble means for each case, domain mean values (horizontally and temporally) after the model spin-up are calculated for each model run. The ensemble means are then used to compare both cases to each other. Error bars represent the standard deviation of the ensemble members.

     In the DST case, mineral dust and soot are available INP for heterogeneous freezing. The setup

requires prescribed concentrations for mineral dust, because no desert, as a requirement for mineral dust emission, is covered by the model domain. Therefore, mineral dust is set to a constant and homogeneously distributed number concentration of $100\,L^{-1}$, adapted to typical atmospheric background concentrations for mineral dust in Europe, when no dust is transported from the Sahara desert towards Europe (Klein et al., 2010; Stanelle et al., 2010). A recent study about seasonal vari-

ations of desert dust confirms this value for a summertime background concentration over Europe (Hande et al., 2014). Soot concentrations are included in the COSMO-ART simulation by prescribed emission rates (Vogel et al., 2009).



## 3   Results

### 3.1   PBAP concentrations

The horizontally distributed bacteria and fungal spore number concentrations at the lowest model layer ($\sim 10\ m$ above ground) are shown in Figure 2 and 3 for the selected case study, to illustrate the order of magnitude of aerosol concentrations that might serve as INP. Bacteria concentrations (Figure 2) over land are in the order of $\sim 10\ L^{-1}$ ($6\ L^{-1}$ and much lower ($< 1\ L^{-1}$) over the ocean. As bacteria emissions are not defined for sea surfaces, land-emitted bacteria can only be transported

there and decrease with distance from the shore due to sedimentation. The emission of bacteria in this study is constant with time, therefore, the amount and distribution of bacteria concentrations are only dependent on transport and removal processes. At surface level, the simulated fungal spore concentrations (Figure 3) are always higher than the bacteria concentrations. Typical fungal spore concentrations over land are between $\sim 10\ L^{-1}$ and $100\ L^{-1}$, which is in agreement with the range

given in other studies (Després et al., 2012, and references therein). In this study, fungal spore concentrations increase towards southern latitudes of the model domain, because the time-dependent emission parameterization is also depending on temperature and specific humidity which are usually higher towards southern latitudes (on the Northern hemisphere). Highest fungal spore concentrations occur in the Italian Po valley, where it is wet and warm together with high amounts of vegetated or

agricultural areas. Horizontal distribution of bacteria and fungal spores in the lowest, surface-most model layer strongly depends on present meteorological conditions, i.e. wind direction and intensity, advection, and emission drivers.

Figure 4 shows a vertical profile of PBAP and INP number concentration. Before calculating the horizontal mean values, a vertical model layer interpolation over equal heights is performed

to compensate the layer deformation due to model topography. Additionally, all data points that are included in the averaged lines are shown in a scatter cloud. This highlights the wide range of PBAP concentrations that are occurring especially in the upper model layers. Due to strong removal processes, the background concentration is very low in the mid-layer of the atmosphere, and thus higher concentrations are mainly given by uplifting events that are stronger than the average.

The vertically distributed average concentrations of bacteria and fungal spores are similar to each other and show a decrease by about three orders of magnitude between the surface level ($\sim 10\ L^{-1}$) and the tropopause layer at $\sim 10\ km$ (Figure 4). The figure additionally shows the vertical distribution of mineral dust INP and PB-INP, which will be described in detail in the following sections.

### 3.2   Diagnostic INP

As a first step, the potential influence of PB-INP on clouds through the immersion freezing of cloud droplets is estimated by analyzing the diagnostic INP concentrations. The term "diagnostic" refers to simulations without feedback between PB-INP and clouds (see Section 3.3). The concentration of



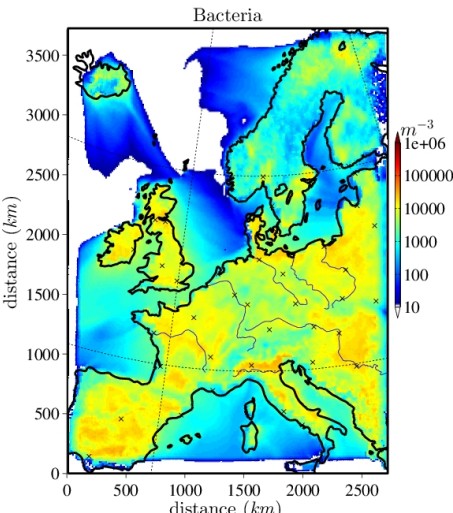

**Figure 2.** Bacteria concentration at surface level

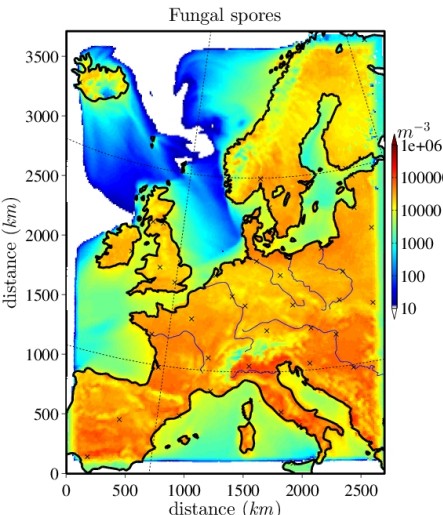

**Figure 3.** Fungal spore concentration at surface level

diagnostic PB-INP ($N_{bioIN}$) at each grid point of the model domain results from the parameterization for ice nucleation by PBAP applied to the temperature and concentration of PBAP ($N_f$) in the

current grid box (Eq. 6).





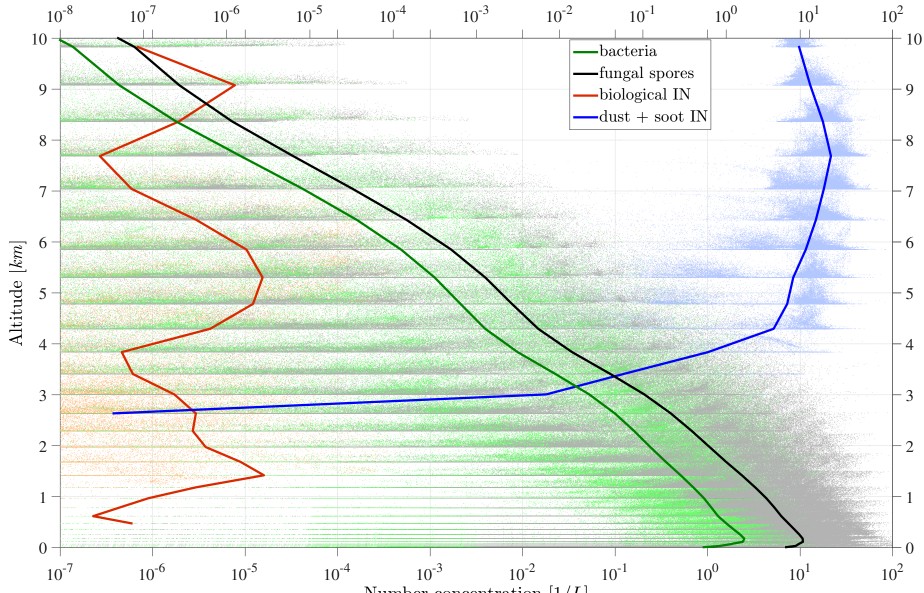

**Figure 4.** Vertical profile domain-mean concentrations of bacteria, fungal spores, total PB-INP, as well as background dust and soot INP concentrations for the case study in July 2010. Each scatter cloud represents the entirety of the original data.

Influences of PB-INP on cloud properties by immersion freezing require supercooled liquid cloud droplets. Therefore, all results of INP concentrations refer to in-cloud conditions. Horizontal averaged PBAP and INP concentrations, resulting in a vertical profile (Figure 4), can represent their abundance at cloud altitudes and thus potential impacts on cloud ice crystal concentration.

Total PB-INP shows highest number concentrations at around $5\ km$ above the surface. This results from highest activity of the most efficient types of PB-INP, i.e. bacteria. Additionally, two small maxima at (i) $1.6\ km$ and (ii) $9\ km$ occur due to (ii) less active types of PB-INP, i.e. *Cladosporium* fungal spores and (i) slightly more active types of PB-INP with lower concentrations than the bacteria, i.e. *Mortierella* fungal spores. A detailed analysis is shown in the following. A decrease of

PB-INP above the topmost peak at $9\ km$ towards higher model levels is caused by transition from heterogeneous to homogeneous ice nucleation. For comparison, the other INP used in the model, mineral dust and soot, are also included in Figure 4. While dust INP reach their highest concentrations at $7.5\ km$, the concentrations above $4\ km$ are already within one order of magnitude of the maximum value. Below a height of $4\ km$, the temperature criterion for heterogeneous freezing of

mineral dust INP is rarely met. Hence, below $3\ km$ only PB-INP are activated.





In the following part, diagnostic INP are shown in 2D-histograms for a more detailed view on the distribution of their concentration. Figures 5 and 6 show the 2D-histograms of bacteria INP and fungal spore INP with their concentration versus grid box temperature. Note that the model level thickness decreases with height which results in higher temperatures occurring more often. A 2D-

histogram avoids calculating mean or median values, which would be dominated by large regions with many low INP concentrations. The color of each box represents the frequency of occurrence of INP at a certain concentration and temperature.

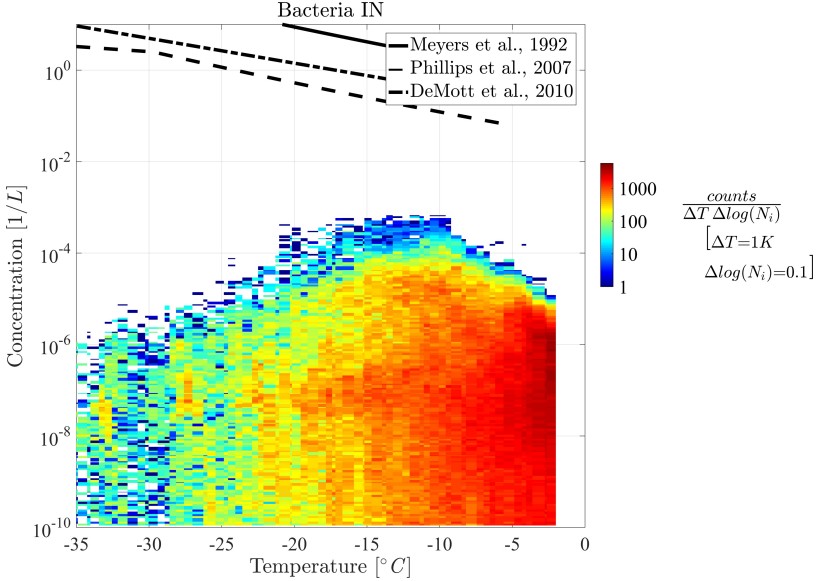

**Figure 5.** 2D-histogram of concentration vs. temperature for bacteria INP.

Both histograms show that maximal INP concentrations clearly decrease towards lower temperatures, but also slightly towards the upper temperature threshold. Thus, highest INP concentrations

of $10^{-3}$ $L^{-1}$ occur at $-9°C$ in case of bacteria INP and at $-5°C$ in case of fungal spore INP. Additionally, the abundance of INP decreases towards lower temperatures, which causes in Figure 4 a decrease in mean concentration (solid line). This effect interferes with a larger extend of the model levels at higher altitudes and hence less data points at low temperatures (less dense scatter cloud in Figure 4). However, decreasing PB-INP concentrations at low temperatures are mainly caused by a

lack of ice nucleation active PBAP. Therefore, both, bacteria INP and fungal spore INP, decrease by about two to three orders of magnitude between $-10°C$ and $-35°C$. This temperature region can be called "PBAP limited".

At temperatures above $-10°C$, the frequency of occurrence of concentration values differs slightly between both types of PB-INP. The maximum concentration of bacteria INP decreases by about two



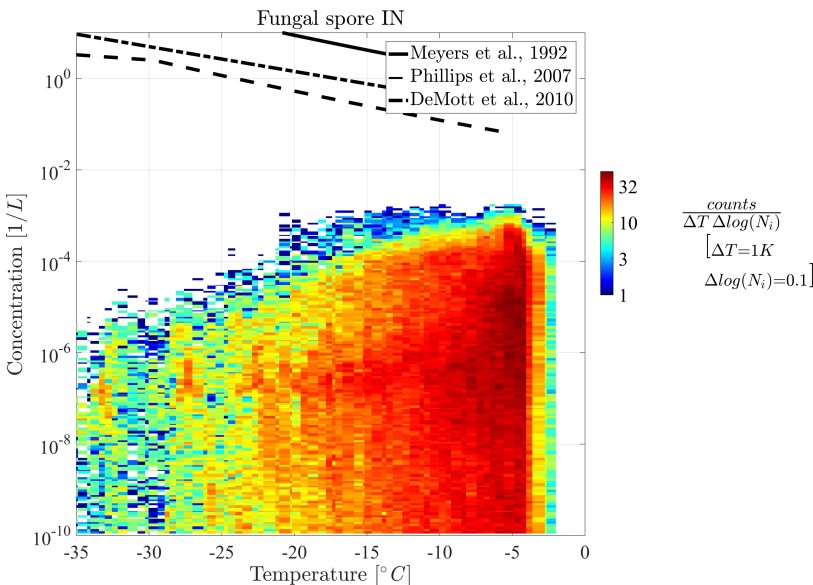

**Figure 6.** 2D-histogram of concentration vs. temperature for both fungal spore INP.

orders of magnitude between $-9°C$ and $-2°C$. At $-2°C$, a sharp edge occurs, because the parameterization for ice nucleation is limited to temperatures below $-2°C$. But still $3 \times 10^{-5}\ L^{-1}$ of bacteria can act as INP at this temperature threshold. Together with high bacteria concentrations, which appear frequently at this temperature, a sharp edge between high and very abundant bacteria INP concentrations and zero INP occurs. In contrast, the concentrations of fungal spore INP decrease to a lesser degree and in a narrow temperature range (between $-5°C$ and $-2°C$). A different mathematical description of heterogeneous ice nucleation with a continuously decreasing slope diminishes the sharp edge at the maximum temperature threshold. For both types of PB-INP analyzed here, bacteria INP and fungal spore INP, concentrations decrease towards their upper temperature threshold, because the parameterization for heterogeneous ice nucleation decreases stronger than its PBAP concentrations increase. This region can be described as "temperature limited", because both parameterizations for heterogeneous ice nucleation are highly temperature-dependent.

### 3.3 Prognostic INP

So called "prognostic" INP are able to effect cloud ice crystal number concentration ($N_i$) and mass mixing ratio within the model. Thus, many different cloud processes, i.e. riming, aggregation to snow, ice multiplication, or melting can be affected. In addition to heterogeneous ice nucleation, other microphysical processes can increase $N_i$, i.e. homogeneous ice nucleation or secondary ice multiplication.





In order to quantify the total effect of PB-INP on cloud properties, vertical profiles of the ensemble means are calculated and compared in the further analysis (Sect. 2.4).

The vertical profile of average prognostic in-cloud INP concentrations ($N_{IN}$) in Figure (7) shows two stages. Below $4\,km$ only PB-INP are active and thus only the case BIO gives prognostic INP concentrations here, which increase with height. Above $4\,km$, concentrations are higher by several orders of magnitude, because other INP, mainly mineral dust, are active too. Here, both cases are very similar to each other, because PB-INP only cause a minor change to these high INP concentrations.

Error bars give the standard deviation of the ensemble members. As the pure PB-INP below a height of $3\,km$ are a few orders of magnitude lower than total INP concentrations at around $8\,km$, the differences between both cases ($\Delta N_{IN}$) reflect this wide range too. Values of $\Delta N_{IN}$ are a few orders of magnitude lower below $3\,km$ height than above $4\,km$ height. The latter has a maximum around $100\,m^{-3}$, which is even larger than the PB-INP concentration as an initial perturbation of the

simulation. Taking the variations between the ensembles into account, high values of $\Delta N_{IN}$ above $4\,km$ are not necessarily caused by this perturbation.

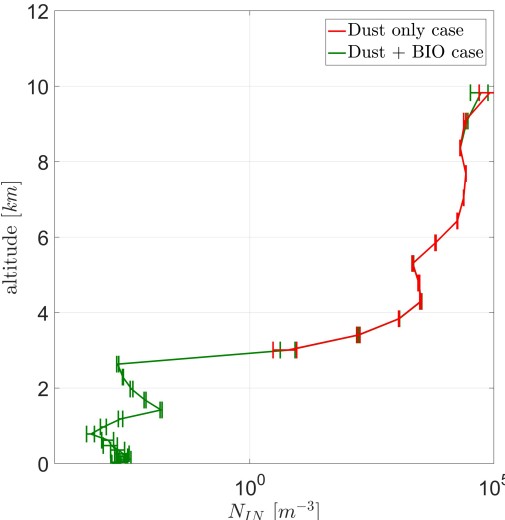

**Figure 7.** Vertical profile for total prognostic in-cloud INP ($N_{IN}$) for a case with and without PB-INP (cases BIO and DST).

With the current model setup, vertical profiles of average ice crystals ($N_i$) for both cases show a maximum concentration at an altitude of $\sim 8\,km$ (Figure 8) of about $1 \times 10^5\,m^{-3}$. Above $9\,km$, $N_i$ is reduced with height, as less ice clouds are present and the tropopause is reached. Between $1\,km$

and $9\,km$, $N_i$ steadily increases with increasing height. Below a height of $1\,km$, $N_i$ is very low,





because melting reduces its number in many regions of the model domain. Except this low area, the standard deviation of the ensemble members are below 5%.

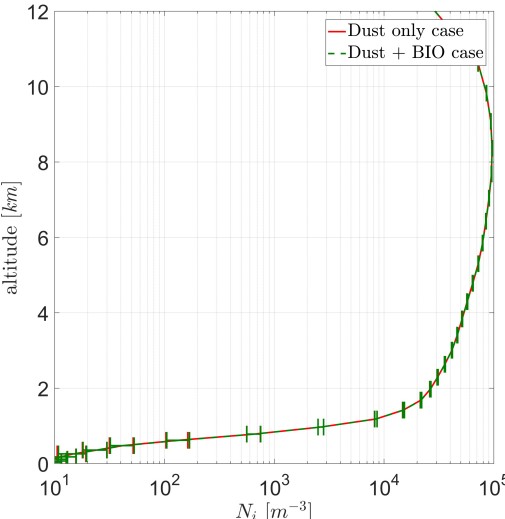

**Figure 8.** Vertical profile of ice crystal number concentration ($N_i$) averaged over the entire domain for both cases (DST and BIO).

As both cases (DST and BIO) are almost equal to each other on average, the difference in $N_i$ between both cases (BIO-DST) is given in Figure 9. Positive values indicate that INP concentrations are higher in case BIO. It shows that mean $N_i$ is almost always enhanced due to PB-INP, but not significantly. Although $\Delta N_i$ is mainly positive, wide error ranges suggest that differences could also be caused by ensemble variations. Nevertheless, largest differences in the ice crystal concentration ($\Delta N_i$) are around $100\ m^{-3}$ and thus similar to differences in INP, but both are well below the error bars.

Although PB-INP on average over the model domain have no significant effect on the ice crystal concentration, they can have some effect under certain conditions, like shallow mixed-phase clouds (i.e. Altostratus) with an cloud top temperature above $-15°C$. In the selected case shown here, an area over Island, the cloud mainly consists of liquid droplets with almost no ice crystals. When focusing on these conditions together with reasonably large PBAP concentrations, PB-INP can create some ice nucleation and thus increase the vertical profile of $N_i$ (Figure 10). However, the number of ice crystals is still very low and the structure of the cloud is probably unchanged, as shown by the vertical profile of $N_c$ which is remains without changes for case "BIO" compared to case "CTL". Hence, PB-INP can influence the cloud ice phase, but in a very limited manner.





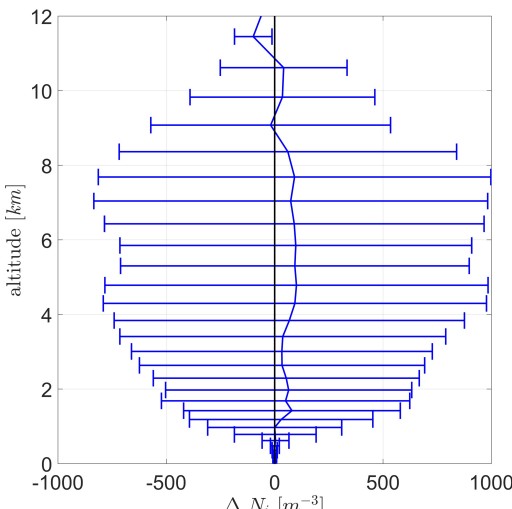

**Figure 9.** Vertical profile of the difference of the ice crystal number concentration ($\Delta N_i$) between the cases BIO minus DST with the same configuration for averaging as Figure 8.

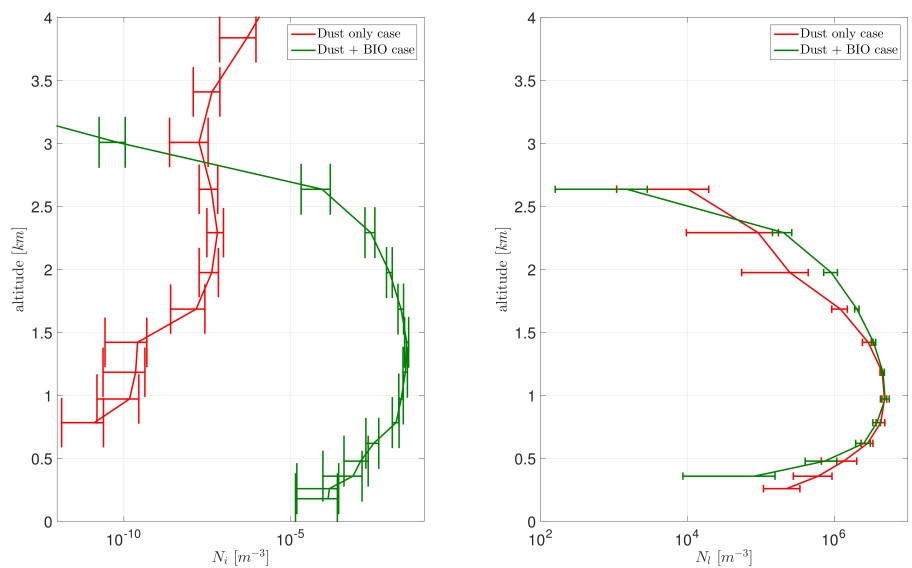

**Figure 10.** Vertical profile of the ice crystal number concentration (left, $N_i$) and the cloud droplet number (right, $N_l$) for the cases DST and BIO for a exemplary section over Iceland.





## 4   Conclusions

PBAP have been implemented into the regional model COSMO-ART by means of individual emission and freezing parameterizations to study their role as INP for mixed-phase clouds. Compared to previous studies on a global scale (Hoose et al., 2010b), a regional model can utilize its higher resolution to observe potentially large temporal and spatial variations in PBAP concentration, which might be able to show a potential influence on the cloud ice phase more clearly.

For the simulation of PBAP concentrations, validated and previously tested emission parameterizations from the literature have been used (Burrows et al., 2009; Hummel et al., 2015). Still, the emission fluxes and atmospheric concentrations hold substantial uncertainties due to unknown or unresolved drivers in the PBAP release mechanism. In the case of bacteria, the emission function is evaluated on a global scale (Burrows et al., 2009), which might not be completely suitable to capture variations from regional scale simulations. Our simulations predict higher near-surface

concentrations for fungal spores than for bacteria. Within the range of uncertainty, the average near-surface concentrations of $\sim 10\ L^{-1}$ shown here are in the same order of magnitude than results from previous modeling studies (Hoose et al., 2010b; Després et al., 2012). PBAP concentrations decrease rapidly with height and are only present in much smaller amounts, on average two orders

of magnitude lower within mixed-phase clouds than near-surface. This reduction is mainly due to an effective washout process and sedimentation taking place for bacteria and fungal spores. Also the spread in number concentrations increases with higher altitude, which makes it difficult to identify an domain-average influence of PB-INP on clouds. However, not much is known about the interstitial and cloud-borne concentrations of PBAP or biological components. Investigations in this research

field would be very valuable for further studies about PB-INP.

PBAP have diverse properties. So far only a small fraction is discovered to be ice-nucleation active and can therefore act as PB-INP. The share of PB-INP within the total PBAP is one of the largest uncertainties (represented by $\varepsilon$) of this study. It is also suggested that PB-INP can break up or burst and spread their ice-nucleation active sites among many fragments (Diehl et al., 2002; Pummer et al.,

2012, 2013). These processes have only been investigated on the microscale, so their contribution on the global scale are currently impossible to be even roughly estimated. Additionally, the temperature threshold that defines the highest temperature at which a certain PB-INP is ice-nucleation active has been implemented into the deterministic parameterization used here. At this threshold, high PB-INP concentrations have the largest frequency of occurrence and a shift of the threshold might be

relevant for influencing the INP concentrations. However, we do not expect that the results of this study, in particular the impact of PB-INP on the ice crystal concentration, changes substantially due to a change in the temperature threshold. The threshold is often only defined by a few data points within the measurements that are underlying this study, so it would be interesting for future studies to also measure the freezing behavior of PB-INP at high temperatures.





To distinguish significant variations in the cloud properties from model variations when comparing both cases (BIO and DST), an ensemble of simulations is performed. Compared to ice crystals, the number concentration of PB-INP is much lower and therefore PB-INP are not able to significantly affect the average state of the ice phase. Even at lower parts of a cloud, with $0° > T > -15°C$ and where no non-biological INP are allowed to be active, ice crystals from upper layers of the cloud fre-

quently perturb the lower layers due to sedimentation, and the changes in ice crystal concentrations in the simulations with PB-INP are smaller than the standard deviation in the ensemble. However, this study shows that PB-INP can create few ice crystals if a cloud with $0° > T > -15°C$ is not perturbed by falling ice crystals, so it only contains liquid droplets. Even in this situation, the cloud properties do not chance substantially. Hence, and most frequently within this simulation, PB-INP

have no effect on the ice crystal concentration.

*Acknowledgements.* The authors wish to thank Marco Paukert and Max Bangert for their help and support with the model, and also Isabelle Steinke and Romy Fösig for their vulnerable discussions. The first author wants to acknowledge Frank Schwarz and Romy Fösig for technical assistance. We acknowledge support by the Deutsche Forschungsgemeinschaft and the Open Access Publishing Fund of the Karlsruhe Institute of Tech-

nology. This research was funded by the Helmholtz Association through the Helmholtz Climate Initiative REK-LIM and the President's Initiative and Networking Fund, and by DFG through project HO 4612/1-1 (FOR 1525 INUIT). The AIDA ice nucleation experiments were funded by the German Science Foundation (DFG) through the project BIOCLOUDS (MO 668/2-1). J. Fröhlich-Nowoisky and B.G. Pummer acknowledge support from the Deutsche Forschungsgemeinschaft (DFG FR3641/1-2, FOR 1525 INUIT).





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
