# Peer review of "Simulating the Influence of Primary Biological Aerosol Particles on Clouds by Heterogeneous Ice Nucleation"

_Atmospheric Chemistry and Physics, 2018_

## Referee Comment (RC1) · Anonymous Referee #1 · 21 Mar 2018

In the manuscript " Simulating the Influence of Primary Biological Aerosol Particles on Clouds by Heterogeneous Ice Nucleation" the authors used regional atmospheric model COSMO-ART to simulate the heterogeneous ice nucleation by PBAPs during a 1-week case study on a domain covering Europe.This is a topic of much recent interest, however linking airborne cells to atmospheric processes is a difficult task, but one that the current authors try to address. So far, there is still a great deal of uncertainty about the role of PB-INP on ice nucleation in mixed-phase clouds and precipitation on the global average.The authors The authors confirmed again that PBAP have no significant influence on the average state of the cloud ice phase at around -10C. The authors only used 1-week case study but if investigate detail, there are still large uncertainties

regarding the difficult problem. The authors should mention or cite the similar result by labortary measurements of rainwater that Lu and collaborators made observation (Lu, et al. 2015. "The diversity and role of biological ice nuclei in rainwater from mountain sites in China." Aerosal and air quality,16: 640–652, 2016 ) thereby illustrating that the representativeness of their conclusions. Generally, I think the topic is great important, the method and data used in this study is sound, the result and conclusions have convinced me, the whole paper is well written.

---

## Author Comment (AC1) · 14 Jun 2018

The authors would like to thank the editor for handling the review of our manuscript. We also wish to thank the reviewer for providing a constructive and timely review.

REVIEWER #1:

The reviewer highlighted the recent interest of the topic, together with the difficulty of the study by M. Hummel et al. of linking airborne biological particle to atmospheric processes. However, the reviewer point out that the authors have addressed the large uncertainty regarding the difficult problem.

[Figure]

The reviewer suggested including the study by Lu et al. ("The diversity and role of biological ice nuclei in rainwater from mountain sites in China." Aerosol and air quality, 16: 640–652, 2016), which shows similar to this study that PBAP have no significant influence on the average state of the cloud ice phase at around -10°C. Lu et al. (2016) used, different from our study, laboratory measurements of rainwater samples to obtain a comparable result.

The authors are considering the results of the study by Lu et al. (2016) in the final manuscript version. The following sentences are added to the second paragraph in the discussion section: "[...] An evidence for biological components in cloud water is the presence of bacteria cells in rainwater samples, focusing on *Pseudomonas* sp. The investigations by Lu et al. (2016) show that *Pseudomonas* sp. are rarely present within rainwater samples collected over eastern China during summer. Similar to findings in this study, Lu et al. (2016) show that the frequency of bacterial INP within the cumulative IN spectrum is very low. However, filtration and heat treatment experiments by Lu et al. (2016) clearly proved the existence of bacterial INP. Further investigations in this research field would be very valuable for further studies about PB-INP."